# Neural Dubber: Dubbing for Videos According to Scripts

**Chenxu Hu[1], Qiao Tian[2], Tingle Li[1,3], Yuping Wang[2], Yuxuan Wang[2], Hang Zhao[1,3]***
[1]IIIS, Tsinghua University    [2]ByteDance    [3]Shanghai Qi Zhi Institute
https://tsinghua-mars-lab.github.io/NeuralDubber/

## Abstract

Dubbing is a post-production process of re-recording actors' dialogues, which is extensively used in filmmaking and video production. It is usually performed manually by professional voice actors who read lines with proper prosody, and in synchronization with the pre-recorded videos. In this work, we propose Neural Dubber, the first neural network model to solve a novel automatic video dubbing (AVD) task: synthesizing human speech synchronized with the given video from the text. Neural Dubber is a multi-modal text-to-speech (TTS) model that utilizes the lip movement in the video to control the prosody of the generated speech. Furthermore, an image-based speaker embedding (ISE) module is developed for the multi-speaker setting, which enables Neural Dubber to generate speech with a reasonable timbre according to the speaker's face. Experiments on the chemistry lecture single-speaker dataset and LRS2 multi-speaker dataset show that Neural Dubber can generate speech audios on par with state-of-the-art TTS models in terms of speech quality. Most importantly, both qualitative and quantitative evaluations show that Neural Dubber can control the prosody of synthesized speech by the video, and generate high-fidelity speech temporally synchronized with the video.

## 1   Introduction

Dubbing is a post-production process of re-recording actors' dialogues in a controlled environment (i.e., a sound studio), which is extensively used in filmmaking and video production. There are two common application scenarios for dubbing. The first one is replacing previous dialogues because poor sound quality is very common for speech recorded on noise location or the scene itself is too challenging to record high-quality audio. The second one is replacing the actor' voices in foreign-language films with those of other performers speaking the audience's language. For example, an English video needs to be dubbed into Chinese if it is shown in China.

In this paper, we mainly focus on the first application scenario, also known as "automated dialogue replacement (ADR)", in which the professional voice actor watches original performance in the pre-recorded video, and re-records each line to match the lip movement with proper prosody such as stress, intonation and rhythm, which allows their speech to be synchronized with the pre-recorded video. In this scenario, the lip motion (viseme) in the video is consistent with the given scripts (phoneme), and the pre-recorded high-definition video can not modified during the ADR process.

While dubbing is an impressive ability of professional voice actors, we aim to achieve this ability computationally. We name this novel task automatic video dubbing (AVD): synthesizing human speech that is temporally synchronized with the given video according to the corresponding text. The main challenges of the task are two-fold: (1) temporal synchronization between synthesized speech

---

*Corresponding to hangzhao@mail.tsinghua.edu.cn

35th Conference on Neural Information Processing Systems (NeurIPS 2021).

and video, i.e., the synthesized speech should be synchronized with the lip movement of the speaker in the given video; (2) the content of the speech should be consistent with the input text.

Text to speech (TTS) is a task closely related to dubbing, which aims at converting given texts into natural and intelligible speech. However, several limitations prevent TTS from being applied in the dubbing problem: 1) TTS is a one-to-many mapping problem (i.e., multiple speech variations can be spoken from the same text) [38], so it is hard to control the variations (e.g., prosody, pitch and duration) in synthesized speech during generation; 2) with only text as input, TTS can not utilize the visual information from the video to control speech synthesis, which greatly limits its applications in dubbing scenarios where the synthesized speech are required to be synchronized with the video.

We introduce Neural Dubber, the first model to solve the AVD task. Neural Dubber is a multi-modal speech synthesis model, which generates high-quality and lip-synced speech from the given text and video. In order to control the duration and prosody of synthesized speech, Neural Dubber works in a non-autoregressive way following [38]. The problem of length mismatch between phoneme sequence and mel-spectrogram sequence in non-autoregressive TTS is usually solved by up-sampling the phoneme sequence according to the predicted phoneme duration. Meanwhile, a phoneme duration predictor is needed, where the ground truth is usually obtained from another model [39, 38] or itself during training [24]. However, due to the natural correspondence between lip movement and text [10], we do not need to get phoneme duration target in advance like previous methods [39, 38, 24]. Instead, we use the text-video aligner which adopts an attention module between the video frames and phonemes, and then upsample the text-video context sequence according to the length ratio of mel-spectrogram sequence and video frame sequence. The text-video aligner not only solves the length mismatch problem, but also allows the lip movement in the video to control the prosody of the generated speech explicitly by the attention between video frames and phonemes.

In the real dubbing scenario, voice actors need to alter the timbre and tone according to different performers in the video. In order to better simulate the real case in the AVD task, we propose the image-based speaker embedding (ISE) module, which aims to synthesize speech with different timbres conditioning on the speakers' face in the multi-speaker setting. To the best of our knowledge, this is the first attempt to predict a speaker embedding from a face image with the goal of generating speech with a reasonable timbre that is consistent with the speaker's facial features (e.g., gender and age). This is achieved by taking advantage of the natural co-occurrence of faces and speech in videos without the supervision of speaker identity. With ISE, Neural Dubber can synthesize speech with a reasonable timbre according to the speaker's face. In other words, Neural Dubber can use different face images to control the timbre of the synthesized speech.

We conduct experiments on the chemistry lecture dataset from Lip2Wav [35] for the single-speaker AVD, and the LRS2 [1] dataset for the multi-speaker AVD. The results of extensive quantitative and qualitative evaluations show that in terms of speech quality, Neural Dubber is on par with state-of-the-art TTS models [51, 41, 38]. Furthermore, Neural Dubber can synthesize speech temporally synchronized with the lip movement in video. In the multi-speaker setting, we demonstrate that the ISE enables Neural Dubber to generate speech with reasonable timbre based on the speaker's face, resulting in Neural Dubber outperforming FastSpeech 2 by a big margin in term of audio quality. We attach some audio files and video clips generated by our model in the supplementary materials.

## 2   Related Work

**Text to Speech.**   Text to Speech (TTS) [3, 41, 51, 39], which aims to synthesize intelligible, natural and high-quality speech from the input text, has seen tremendous progress in recent years. Specifically, the prevalent methods have shifted from concatenative synthesis [19], parametric synthesis [53] to end-to-end neural network-based synthesis [33, 41, 51], where the quality of the synthesized speech is improved by a large margin and is close to that of the human counterpart. The general paradigm of end-to-end neural network-based methods usually first generate the acoustic feature (e.g., mel-spectrogram) from text using encoder-decoder architecture autoregressively [51, 41, 34, 28] or non-autoregressively [39, 38, 24], then reconstruct the waveform signal using vocoder [16, 30, 37, 26, 54]. When it comes to the recent multi-speaker TTS system [20], the speaker embedding is often extracted using speaker verification system, and is fed to the decoder of the TTS system in order to encourage the model to obtain a timbre inclination for the speaker of interest. Different from TTS, Neural

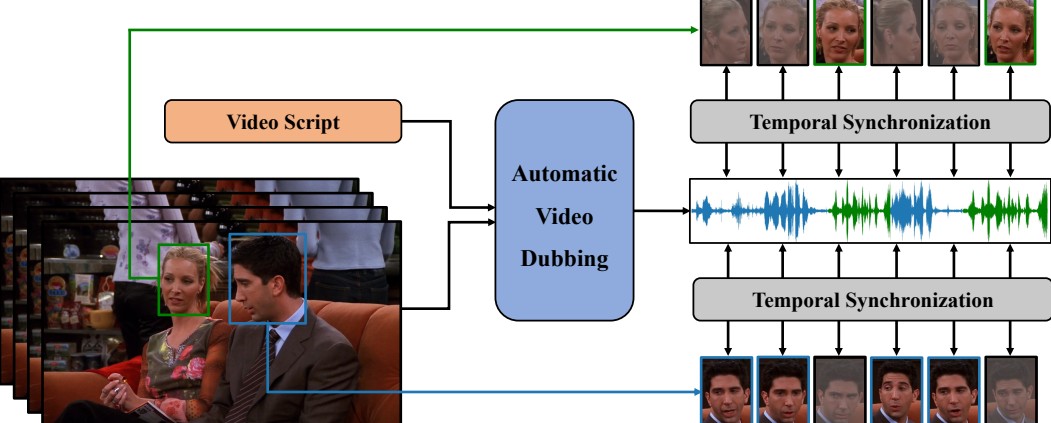

Figure 1: The schematic diagram of the automatic video dubbing (AVD) task. Given the video script and the video as input, the AVD task aims to synthesize speech that is temporally synchronized with the video. This is a scene where two people are talking with each other. The face picture is gray to indicate that the person was not talking at that time.

Dubber is conditioned not only on texts but also on videos, intending to synthesize natural speech given both of them.

**Talking Face Generation.** Talking face generation has a long history in computer vision, ranging from viseme-based models [14, 58] to neural synthesis of 2D [43, 46, 52] or 3D [21, 40, 45] face. Recently neural synthesis approaches have been proposed to generate realistic 2D video of talking heads. Concretely speaking, Chung *et al.* [9] first generates lower face animation using cropped frontal images. After then Zhou *et al.* [57] further disentangles identity from speech using generative adversarial networks (GANs). Wav2Lip [36] tries to explore the problem of visual dubbing, i.e., lip-syncing a talking head video of an arbitrary person to match a target speech segment. From our perspective, however, such methods can not generate high-fidelity face and lip given speech, spawning the results are of low resolution and look uncanny sometimes. Besides, audios in most talking face pipelines need to be prepared in advance, thus, strictly speaking, this does not belong to dubbing (re-recording)[2], but to the face synchronization while given audio. In contrast to the aforementioned works, Neural Dubber is not required to prepare audio beforehand and modify the lip motion, but generates speech audio synchronized with the video from scripts.

**Lip to Speech Synthesis.** Given a video, the lip to speech task aims at synthesizing the corresponding speech audio by directly judging from the lip motion. While the conventional method [22] exploits the visual features extracted from active appearance models, recent end-to-end methods have also shed some light on it. In particular, Vid2Speech [15] and Lipper [27] generate low-dimensional linear predictive coding features to synthesize speech in the constrained scene. Vougioukas *et al.* [49] using the GANs-based method to exert for quality gains. Lip2Wav [35] has achieved promising results in real-life speaker-dependent scenarios, but it is still somewhat incongruous and prone to collapse in the multi-speaker setting. This is possibly because the word error rate in lip reading task [2, 4, 10, 12] is still high, let alone the lip to speech synthesis. In Neural Dubber, the textual information is provided, allowing us to concentrate more on the alignment between the phoneme and lip motion in video, instead of decoding speech from lip motion directly.

## 3 Method

In this section, we first introduce the novel automatic video dubbing (AVD) task; we then describe the overall architecture of our proposed Neural Dubber; finally we detail the main components in Neural Dubber.

---

[2] https://en.wikipedia.org/wiki/Dubbing_(filmmaking)

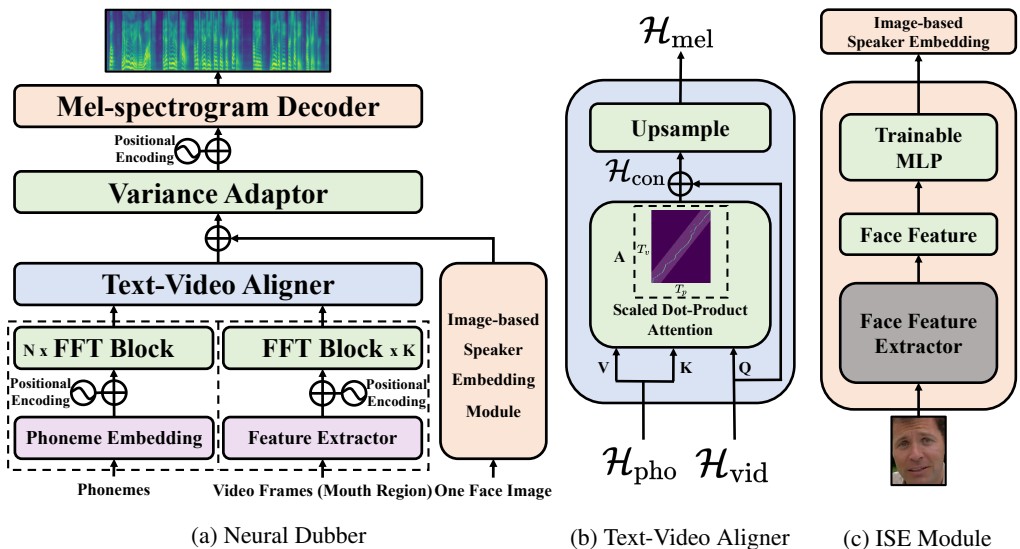

Figure 2: The architecture of Neural Dubber.

### 3.1 Automatic Video Dubbing

Given a sentence $T$ and a corresponding video clip (without audio) $V$, the goal of automatic video dubbing (AVD) is to synthesize natural and intelligible speech $S$ whose content is consistent with $T$, and whose prosody is synchronized with the lip movement of the active speaker in the video $V$. Compared to the traditional speech synthesis task which only generates natural and intelligible speech $S$ given the sentence $T$, AVD task is more difficult due to the synchronization requirement.

### 3.2 Neural Dubber

#### 3.2.1 Design Overview

Our Neural Dubber aims to solve the AVD task. Concretely, we formulate the problem as follows: given a phoneme sequence $S_p = \{P_1, P_2, \ldots, P_{Tp}\}$ and a video frame sequence $S_v = \{I_1, I_2, \ldots, I_{Tv}\}$, we need to predict a target mel-spectrogram sequence $S_m = \{Y_1, Y_2, \ldots, Y_{Tm}\}$.

The overall model architecture of Neural Dubber is shown in Figure 2. First, we apply a phoneme encoder $f_p$ and a video encoder $f_v$ to process the phonemes and images respectively. Note that the images we feed to the video encoder only contain mouth region of the speaker following [10, 32, 42]. We use $S_v^m$ to represent these images. After the encoding, raw phonemes turn into a sequence of hidden representations $\mathcal{H}_{pho} = f_p(S_p) \in \mathbb{R}^{T_p \times d}$ while images of mouth region turn into a sequence of hidden representations $\mathcal{H}_{vid} = f_v(S_v^m) \in \mathbb{R}^{T_v \times d}$. Then we feed $\mathcal{H}_{pho}$ and $\mathcal{H}_{vid}$ into the text-video aligner (which will be described in Section 3.2.3) and get the expanded sequence $\mathcal{H}_{mel} \in \mathbb{R}^{T_m \times d}$ with the same length as the target mel-spectrogram sequence $S_m$. Meanwhile, a face image randomly selected from the video frames is input into image-based speaker embedding (ISE) module (which will be described in Section 3.2.4) to generate a image-based speaker embedding. We add $\mathcal{H}_{mel}$ and ISE together and feed them into the variance adaptor to add some variance information (e.g., pitch and energy). Finally, we use the mel-spectrogram decoder to convert the adapted hidden sequence into mel-spectrogram sequence following [39]. Different from FastSpeech 2 [38], our variance adaptor consists of pitch and energy predictors without duration predictor, because we solve the problem of length mismatch between the phoneme and mel-spectrogram sequence in the text-video aligner and the input of variance adaptor is as long as the mel-spectrogram sequence.

#### 3.2.2 Phoneme and Video Encoders

The phoneme encoder and video encoder are shown in Figure 2a, which are enclosed in a dashed box. The function of the phoneme encoder and video encoder is to transform the original phoneme and image sequences into hidden representation sequences which contain high-level semantics. The phoneme encoder we use is similar to that in FastSpeech [39], which consists of an embedding layer

and N Feed-Forward Transformer (FFT) blocks. The video encoder consists of a feature extractor and K FFT blocks. The feature extractor is a CNN backbone that generates feature representation for every input mouth image. And then we use the FFT blocks to capture the dynamics of the mouth region because FFT is based on self-attention [48] and 1D convolution where self-attention and 1D convolution are suitable for capturing long-term and short-term dynamics respectively.

### 3.2.3 Text-Video Aligner

The most challenging aspect of the AVD task is alignment: (1) the content of the generated speech should come from the input phonemes; (2) the prosody of the generated speech should be aligned with the video in time. So it does not make sense to produce speech solely from phonemes, nor video. In our design, the text-video aligner (Figure 2b) aims to find the correspondence between text and lip movement first, so that synchronized speech can be generated in the later stage.

In the text-video aligner, an attention-based module learns the alignment between the phoneme sequence and the video frame sequence, and produces the text-video context sequence. Then an upsampling operation is performed to change the length of the text-video context sequence $\mathcal{H}_{con}$ from $T_v$ to $T_m$.

In practice, we adopt the popular Scaled Dot-Product Attention [48] as the attention module, where $\mathcal{H}_{vid}$ is used as the query, and $\mathcal{H}_{pho}$ is used as both the key and the value.

$$\text{Attention}(Q, K, V) = \text{Attention}\left(\mathcal{H}_{vid}, \mathcal{H}_{pho}, \mathcal{H}_{pho}\right) \tag{1}$$

$$= \text{Softmax}\left(\frac{\mathcal{H}_{vid}\mathcal{H}_{pho}^T}{\sqrt{d}}\right)\mathcal{H}_{pho} \tag{2}$$

$$= A\mathcal{H}_{pho} \in \mathbb{R}^{T_v \times d}, \tag{3}$$

where $A$ is the matrix of attention weights. After the attention module, we get the text-video context sequence, i.e., the expanded sequence of phoneme hidden representation by linear combination. We use a residual connection [18] to add the $\mathcal{H}_{vid}$ for efficient training. However, we use a dropout layer with a large dropout rate to prevent mel-spectrograms from being generated directly from visual information. The attention weight $A$ obtained after softmax is the main determinant of the speed and prosody of the synthesized speech like the attention weight between spectrograms and phonemes in [51, 41, 28]. The sequence of video hidden representations is used as the query. As a consequence, the attention weight is controlled by the video explicitly, and the temporal alignment between video frames and phonemes is achieved. The obtained monotonic alignment between video frames and phonemes contributes to the synchronization between the synthesized speech and the video on fine-grained (phoneme) level.

There is a natural temporal correspondence between the speech audio and the video. In other words, once the alignment between video frames and phonemes is achieved, the alignment between mel-spectrogram frames and phonemes can be obtained. In practice, the length of a mel-spectrograms sequence is $n$ times that of a video frame sequence. We denote the $n$ as

$$n = \frac{T_{mel}}{T_v} = \frac{sr/hs}{FPS} \in \mathbb{N}^+, \tag{4}$$

where sr denotes the sampling rate of the audio and hs denotes hop size set when transforming the raw waveform into mel-spectrograms. We upsample the text-video context sequence $\mathcal{H}_{con}$ to $\mathcal{H}_{mel}$ with scale factor is $n$. Actually, we use the upsampling method with nearest mode.

$$\mathcal{H}_{con} = \{C_1, C_2, \ldots, C_{Tv}\} \in \mathbb{R}^{T_v \times d} \tag{5}$$

$$\mathcal{H}_{mel} = \text{Upsample}\left(\mathcal{H}_{con}, n\right) \in \mathbb{R}^{T_m \times d} \tag{6}$$

After that, the length of the text-video context sequence is expanded to that of the mel-spectrograms sequence. Thus, the length mismatch problem between the phoneme and mel-spectrograms sequence is solved without the supervision of fine grained alignment between phonemes and spectrograms. Because of the attention between video frames and phonemes, the speed and part of prosody of synthesized speech are controlled by input video explicitly, which makes the synthesized speech and input video well synchronized.

**Monotonic Alignment Constraint**  In text to speech (TTS) task, the monotonic and diagonal alignments in the attention weights between text and speech are important to ensure the quality of synthesized speech [51, 41, 44, 6]. In Neural Dubber, a multi-modal TTS model, the monotonic and diagonal alignments between video frames and phonemes are also critical. So we adopt a diagonal constraint on the attention weights to guide the text-video attention module to learn right alignments following [6]. We formulate the diagonal attention rate $r$ as

$$r = \frac{\sum_{s=1}^{T_m} \sum_{t=\max(ks-b,1)}^{\min(ks+b,T_p)} A_{s,t}}{T_m},$$  (7)

where $k = \frac{T_p}{T_m}$, $b$ is a hyperparameter for bandwidth of the diagonal area. We add the diagonal constraint loss which is defined as $L_{DC} = -r$ to our final loss for better alignments.

### 3.2.4  Image-based Speaker Embedding Module

How much can we infer about the way people speak from their appearances? In the real dubbing scenario, voice actors need to alter the timbre according to different performers. In order to better simulate the real case in AVD task, we aim to synthesize speech with different timbre conditioning on the speakers' faces in multi-speaker setting. There have been many works [29, 23, 13] researching the correlation between voice and speakers' face recently, but none of them learn the joint speaker-face embeddings to solve the multi-speaker text to speech task. In this work, we propose image-based speaker embedding (ISE) module (Figure 2c), a new multi-modal speaker embedding module, generates an embedding that encapsulates the characteristics of the speaker's voice from an image of his/her face. The ISE module is trained with other components of Neural Dubber from scratch in a self-supervised manner, utilizing the natural co-occurrence of faces and speech audio in videos, but without the supervision of speaker identity. We randomly select a face image $I_i^f$ from $S_v^f = \left\{ I_1^f, I_2^f, \ldots, I_{Tv}^f \right\}$, and obtain a high-level face feature by feeding the selected face image into a pre-trained and fixed face recognition network [31, 5]. Then we feed the face feature to a trainable MLP and gain the ISE. The predicted ISE is directly broadcasted and added to $\mathcal{H}_{mel}$ so as to control the timbre of synthesized speech. Our model learns face-voice correlations which allow it to produce speech that coincides with various voice attributes of the speakers (e.g., gender and age) inferred from their faces.

## 4  Experiments and Results

### 4.1  Datasets

**Single-speaker Dataset**  In the single-speaker setting, we conduct experiments on the chemistry lecture dataset from Lip2Wav [35]. With a large vocabulary size and a lot of head movements, the dataset is originally used for the unconstrained single-speaker lip to speech synthesis. To make it fit the AVD task, we collect the official transcripts from YouTube. We need corresponding sentence-level text and audio clips for training, so we segment the long videos into sentence-level clips according to the start and end timestamp of each sentence in the transcripts. Some segmented sentence-level video clips contain frames that only capture the PowerPoint but not lecturer face which can not be used for training. So we conduct data cleaning to remove them. Finally, the dataset contains 6,640 samples, with the total video length of approximately 9 hours. We randomly split the dataset into 3 sets: 6240 samples for training, 200 samples for validation, and 200 samples for testing. In the following subsections, we refer to this dataset as chem for short.

**Multi-speaker Dataset**  In multi-speaker setting, we conduct experiments on the LRS2 [1] dataset, which consists of thousands of sentences spoken by various speakers on BBC channels. This dataset suits the AVD task well, because each sample includes both the text and video pair. Note that we only train on the training set of the LRS2 dataset, which only contains data of approximately 29 hours. Compared to other multi-speaker speech synthesis datasets [55], this dataset is quite small for multi-speaker speech generation and does not provide the speaker identity for each sample. The ISE module aids Neural Dubber in solving these problems.

## 4.2 Data Pre-processing

The video frames are sampled at 25 FPS. We detect and crop the face from the video frames using $S^3FD$ [56] face detection following [35]. The images input to the video encoder are resized to 96 x 96 in dimension, which only cover the mouth region of the face, as shown in Figure 2a. The face image input to the ISE module is 224 x 224 in dimension and covers the whole face of the speaker. In order to alleviate the mispronunciation problem, we convert the text sequences into the phoneme sequences [3, 28, 38] with an open-source grapheme-to-phoneme tool. For the speech audio, we transform the raw waveform into mel-spectrograms following [41]. The frame size and hop size are set to 640 samples (40 ms) and 160 samples (10 ms) with respect to the sample rate 16 kHz.

## 4.3 Model Configuration

**Neural Dubber**    Our Neural Dubber consists of 4 feed-forward Transformer (FFT) blocks [39] in the phoneme encoder, the mel-spectrogram decoder, and 2 FFT blocks in the video encoder. The feature extractor in the video encoder is the ResNet18 [18] except for the first 2D convolution layer being replaced by 3D convolutions [32]. The variance adaptor contains pitch predictor and energy predictor. The configurations of the FFT block, the mel-spectrogram decoder, the pitch predictor and the energy predictor are the same as those in FastSpeech 2 [38]. In the text-video aligner, the hidden size of the scaled dot-product attention is set to 256, the number $n$ of the repeat operation is set to 4 according to Equation (4). In the ISE module, the face feature extractor we use is a pre-trained and fixed ResNet50 trained on the VGGFace2 [5] dataset. The face feature is a 4096-D feature that is extracted from the penultimate layer (i.e., one layer prior to the classification layer) of the network.

**Baseline**    Since automatic video dubbing is a new task that we propose, none of the previous works focused on solving this task. So we propose a baseline model based on the Tacotron [51] system with some modifications which make it fit to the new AVD task. We call this baseline model **Video-based Tacotron**. In order to make use of the information in video, we concatenate the spectrogram frames with the corresponding hidden representation of video frames, and use it as the decoder input:

$$Y_{i-1}^{'} = Y_{i-1} \oplus \mathcal{H}_{vid}^{\lceil \frac{i}{n} \rceil} \quad , \tag{8}$$

where $Y_{i-1}^{'}$ is the decoder input, $\oplus$ represents the concatenation operation, $\mathcal{H}_{vid}$ is the hidden representation of video frames, which is obtained in the same way as in Neural Dubber described in the Section 3.2.1 and $n$ is same as that in Equation (4). The Video-based Tacotron implementation is based on an open-source Tacotron repository [3] where the attention is replaced with the location-sensitive attention [7] according to [41] for better results. We set the reduction factor $r$ to 2 and change the vocoder to Parallel WaveGAN [54] for fair comparison.

## 4.4 Training and Inference

We train Neural Dubber on 1 NVIDIA V100 GPU. We use the Adam optimizer [25] with $\beta_1 = 0.9$, $\beta_2 = 0.98$, $\varepsilon = 10^{-9}$ and follow the same learning rate schedule in [48]. Our model is optimized with the loss similar to that in [38]. We set the batchsize to 18 and 24 on chem dataset and LRS2 dataset respectively. It takes 200k/300k steps for training until convergence on the chem/LRS2 dataset. In this work, we use Parallel WaveGAN [54] as the vocoder to transform the generated mel-spectrograms into audio samples. We train two Parallel WaveGAN vocoders on the training set of chem dataset and LRS2 dataset respectively, following an open-source implementation [4]. Each Parallel WaveGan vocoder is trained on 1 NVIDIA V100 GPU for 1000K steps. In the inference process, the output mel-spectrograms of Neural Dubber are transformed into audio samples using the pre-trained Parallel WaveGAN.

## 4.5 Evaluation

### 4.5.1 Metrics

Since the AVD task aims to synthesize human speech synchronized with the video from text, the audio quality and the audio-visual synchronization (av sync) are the important evaluation criteria.

---

[3]https://github.com/fatchord/WaveRNN
[4]https://github.com/kan-bayashi/ParallelWaveGAN

**Human Evaluation**   We conduct the mean opinion score (MOS) [8] evaluation on the test set to measure the audio quality and the av sync. We randomly select 30 video clips from the test set, where each video clip is scored by at least 20 raters, who are all native English speakers. We overlay the synthesized speech on the original video before showing it to the rater, following [35]. The text and the video are consistent among different systems, so that all raters only examine the audio quality and the av sync without other interference factors. For each video clip, the raters are asked to rate scores of 1-5 from bad to excellent (higher score indicates better quality) on the audio quality and the av sync, respectively. We perform the MOS evaluation on Amazon Mechanical Turk (MTurk).

**Quantitative Evaluation**   In order to measure the synchronization between the generated speech and the video quantitatively, we use the pre-trained SyncNet [11], which is publicly available[5] following [35]. The method can explicitly test for synchronization between speech audio and lip movements in unconstrained videos in the wild [11, 36]. We adopt two metrics: Lip Sync Error - Distance (LSE-D) and Lip Sync Error - Confidence (LSE-C) from Wav2Lip [36]. The two metrics can be automatically calculated by the pre-trained SyncNet model. LSE-D denotes the minimal distance between the audio and the video features for different offset values. A lower LSE-D means the speech audio and video are more synchronized. LSE-C denotes the confidence that the audio and the video are synchronized with a certain time offset. A lower LSE-C means that some parts of the video are completely out of sync, where the audio and the video are uncorrelated.

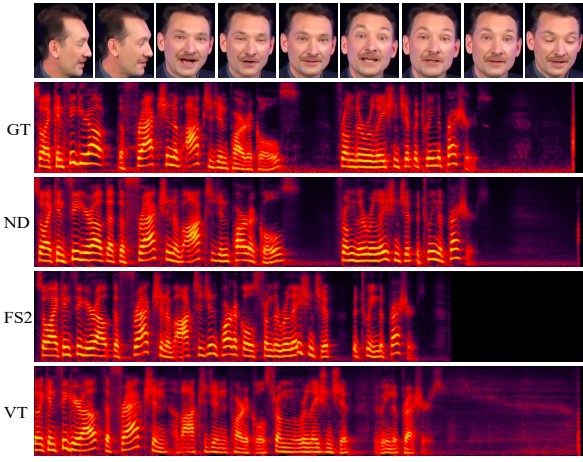

Figure 3: Mel-spectrograms of audios synthesized by some systems: Ground Truth (GT), Neural Dubber (ND), FastSpeech 2 (FS2) and Video-based Tacotron (VT).

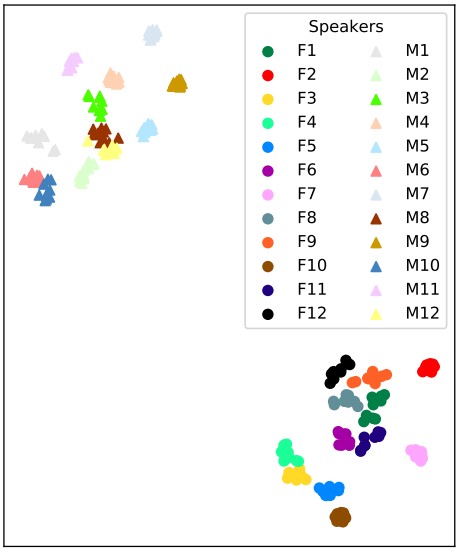

Figure 4: Speaker embedding visualization.

#### 4.5.2   Single-speaker AVD

We first conduct MOS evaluation on the chem single-speaker dataset, to compare the audio quality and the av sync of the video clips generated by Neural Dubber with other systems, including 1) GT, the ground-truth video clips; 2) GT (Mel + PWG), where we first convert the ground-truth audio into mel-spectrograms, and then convert it back to audio using Parallel WaveGAN [54] (PWG); 3) FastSpeech 2 [38] (Mel + PWG); 4) Video-based Tacotron (Mel + PWG). Note that the systems in 2), 3), 4) and Neural Dubber use the same pre-trained Parallel WaveGAN for a fair comparison. In addition, we compare Neural Dubber with those systems on the test set using the LSE-D and LSE-C metrics. The results for single-speaker AVD are shown in Table 1. It can be seen that Neural Dubber can surpass the Video-based Tacotron baseline and is on par with FastSpeech 2 in terms of audio quality, which demonstrates that Neural Dubber can synthesize high-quality speech. Furthermore, in terms of the av sync, Neural Dubber outperforms FastSpeech 2 and Video-based Tacotron by a big margin and matches GT (Mel + PWG) system in both qualitative and quantitative evaluations, which shows that Neural Dubber can control the prosody of speech and generate speech synchronized with the video. For FastSpeech 2 and Video-based Tacotron, the LSE-D is high and the LSE-C is low,

---

[5] https://github.com/joonson/syncnet_python

indicating that they can not generate speech synchronized with the video. We also show a qualitative comparison in Figure 3 which contains mel-spectrograms of audios generated by the above systems. It shows that the prosody of the audio generated by Neural Dubber is closed to that of ground truth recording, i.e., well synchronized with the video.

| Method | Audio Quality | AV Sync | LSE-D ↓ | LSE-C ↑ |
|---|---|---|---|---|
| *GT* | $3.93 \pm 0.08$ | $4.13 \pm 0.07$ | 6.926 | 7.711 |
| *GT (Mel + PWG)* | $3.83 \pm 0.09$ | $4.05 \pm 0.07$ | 7.384 | 6.806 |
| *FastSpeech 2 [38] (Mel + PWG)* | $3.71 \pm 0.08$ | $3.29 \pm 0.09$ | 11.86 | 2.805 |
| *Video-based Tacotron (Mel + PWG)* | $3.55 \pm 0.09$ | $3.03 \pm 0.10$ | 11.79 | 2.231 |
| *Neural Dubber (Mel + PWG)* | $3.74 \pm 0.08$ | $3.91 \pm 0.07$ | 7.212 | 7.037 |

Table 1: The evaluation results for the single-speaker AVD. The subjective metrics for audio quality and av sync are with 95% confidence intervals.

In addition, we compare our method with another baseline [17] which automatically stretches and compresses the audio signal to match the lip movement given an unaligned face sequence and speech audio. We use the speech generated by FastSpeech 2 (Mel + PWG) system, and then align the pre-generated speech with the lip movement according to [17]. However, the quality and naturalness of its synthesized speech is much worse than the pre-generated speech due to challenging alignments. So this baseline is not comparable to our Neural Dubber.

### 4.5.3 Multi-speaker AVD

Similar to Section 4.5.2, we conduct human evaluation and quantitative evaluation on the LRS2 multi-speaker dataset to compare Neural Dubber with other systems in multi-speaker setting. Due to the failure of Video-based Tacotron in single-speaker AVD, we no longer compare our model with it. Note that we can not add a trivial speaker embedding module to FastSpeech 2, because the LRS2 dataset does not contain the speaker identity for each video. So we directly train FastSpeech 2 on the LRS2 dataset without modifications. The results are shown in Table 2. We can see that Neural Dubber outperforms FastSpeech 2 by a significant margin in terms of audio quality, exhibiting the effectiveness of ISE in multi-speaker AVD. The qualitative and quantitative evaluations show that the speech synthesized by Neural Dubber is much better than that of FastSpeech 2 and is on par with the ground truth recordings in terms of synchronization. These results show that Neural Dubber can address the multi-speaker AVD which is more challenging than the single-speaker AVD.

In order to demonstrate that ISE enables Neural Dubber to control the timbre by the input face image, some audio clips are generated by Neural Dubber with the same phoneme sequence and mouth image sequence but different speaker face images as input. We select 12 males and 12 females from the test set of LRS2 dataset for this evaluation. For each person, we chose 10 face images with different head posture, illumination and facial makeup, etc.

We visualize the voice embedding of these audios in Figure 4 by using a pre-trained speaker encoder [50] from an open-source repository[6]. We first use the speaker (voice) encoder to derive a high-level representation, i.e., a 256-D embedding, from an audio, which summarizes the characteristics of the voice in the audio. Then we use t-SNE [47] to visualize the generated embedding. It can be seen that the utterances generated from the images of the same speaker form a tight cluster, and that the cluster representing each speaker is separated from each other. In addition, there is a distinctive discrepancy between the speech synthesized from the face images of different genders. It concludes that Neural Dubber can use the face image to alter the timbre of the generated speech.

### 4.5.4 Comparing with the Lip-motion Based Speech Generation Method

Recently, some works have demonstrated the impressive ability to generate speech directly from the lip motion. However, the quality and intelligibility of the generated speech are relatively poor, and the word error rate (WER) is very high. In this section, we compare with a SOTA lip-motion based speech generation system Lip2Wav [35]. Because Lip2Wav can only generate word-level speech in the multi-speaker setting, we only compare Neural Dubber with Lip2Wav in the single-speaker

---

[6]https://github.com/resemble-ai/Resemblyzer

| Method | Audio Quality | AV Sync | LSE-D ↓ | LSE-C ↑ |
|---|---|---|---|---|
| *GT* | $3.97 \pm 0.09$ | $3.81 \pm 0.10$ | 7.214 | 6.755 |
| *GT (Mel + PWG)* | $3.92 \pm 0.09$ | $3.69 \pm 0.11$ | 7.317 | 6.603 |
| *FastSpeech 2 [38] (Mel + PWG)* | $3.15 \pm 0.14$ | $3.33 \pm 0.10$ | 10.17 | 3.714 |
| *Neural Dubber (Mel + PWG)* | $3.58 \pm 0.13$ | $3.62 \pm 0.09$ | 7.201 | 6.861 |

Table 2: The evaluation results for the multi-speaker AVD. The subjective metrics for audio quality and av sync are with 95% confidence intervals.

setting still on the chem dataset. We use the official GitHub repository[7] to train Lip2Wav on our version of the chemistry lecture dataset. As we mentioned in Section 4.1, the dataset is different from the original one in Lip2Wav. It only contains data of approximately 9 hours, which is much less than the original one (approximately 24 hours). In this experiment, the training and testing sets of Neural Dubber and Lip2Wav are identical, so the results can be compared directly. Following the Lip2Wav paper [35], we use STOI and ESTOI for estimating the intelligibility and PESQ for measuring the quality. In addition, using an out-of-the-box ASR system, we evaluate the speech results using word error rates (WER). In order to eliminate the influence of the ASR system, we also measure the WER for ground truth speech audio. All these metrics are computed on the test dataset.

| Method | STOI ↑ | ESTOI ↑ | PESQ ↑ | WER ↓ |
|---|---|---|---|---|
| Ground Truth | NA | NA | NA | 7.57% |
| Lip2Wav | 0.282 | 0.176 | 1.194 | 72.70% |
| **Neural Dubber (ours)** | **0.467** | **0.308** | **1.250** | **18.01%** |

Table 3: The comparison between Lip2Wav and Neural Dubber for the single-speaker setting on the chemistry lecture dataset.

As the comparison results in Table 3 show, Neural Dubber surpasses Lip2Wav by a big margin in terms of speech quality and intelligibility. Please note that STOI, ESTOI, and PESQ scores of Lip2Wav are lower than those in [35], because the training data we used is much less than theirs. Most importantly, the WER of Neural Dubber is 4x lower than that of Lip2Wav. It shows that Neural Dubber outperforms Lip2Wav significantly in pronunciation accuracy. WER of Lip2Wav is up to 72.70%, indicating that it mispronounces a lot of content, which is unacceptable in the AVD task. Just like it is unacceptable for an actor to always mispronounce the lines. Please note that the WER of Lip2Wav we get is consistent with the results in [35] (see its Table 5). In summary, Neural Dubber far outperforms Lip2Wav in terms of speech intelligibility, quality, and pronunciation accuracy (WER), and is much more suitable for the AVD task.

## 5   Limitations and Societal Impact

When the script is changed to be different from what the speaker is actually saying, our method can only deal with the situation of modifying couple words. In addition, the lip movement of the modified text should be similar to the original lip movement in the video. The facial appearance may lead to timbre ambiguity due to the dataset bias. It might be offensive. Our method can dub videos automatically, which may be useful for filmmaking and video production.

## 6   Conclusion

In this work, we introduce a novel task, automatic video dubbing (AVD), which aims to synthesize human speech synchronized with the given video from text. To solve the AVD task, we propose Neural Dubber, a multi-modal TTS model, which can generate lip-synced mel-spectrograms in parallel. We design several key components including the video encoder, the text-video aligner and the ISE module for Neural Dubber to better solve the task. Our experimental results show that, in terms of speech quality, Neural Dubber is on par with FastSpeech 2 on the chem dataset, even outperforms FastSpeech 2 on the LRS2 dataset due to ISE's help. More importantly, Neural Dubber can synthesize speech temporally synchronized with the video.

---

[7]https://github.com/Rudrabha/Lip2Wav

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
