# OpenReview forum: "Neural Dubber: Dubbing for Videos According to Scripts"
_NeurIPS.cc/2021/Conference — NeurIPS 2021 Poster_

### Official Review · Reviewer_nxuo · 2021-07-16

**Rating:** 6
**Confidence:** 4

**Summary:**

This paper presents "Neural Dubber", a method for dubbing silent videos based on given scripts. It essentially performs lip-synced TTS, with an added component (ISE) that ensures that the generated speech characteristics match the given face image.

Experiments verify that the proposed TTS method does not hurt SOTA TTS performance, while controlling the prosody such that the aligned speech is much more synchronized than other methods compared to. The supplementary video qualitatively demonstrates some compelling results of the proposed method.

**Limitations And Societal Impact:**

IMO this aspect of the paper is missing. Discussing limitations will strengthen the manuscript, not weaken it. In addition, the proposed method could potentially be used for generating fake talking videos, but this depends on how well the method works on scripts different from what is actually being said by the speaker in the video (I assume this will not work well enough to deceive people).

**Main Review:**

**Originality:**

The task of performing end-to-end lip-synced TTS is, as far as I am aware of, a novel one. The seemingly effective method used to solve this task is a well-motivated combination of components well-documented in the literature: per-modality feature extraction and processing using transformers --> attention-based alignment --> joint decoding.

The main weakness of this work is its conspicuously-missing reference to a very related work: "Dynamic Temporal Alignment of Speech to Lips", Halperin et al. (ICASSP'19), which, given an unaligned face sequence and audio track, automatically stretches and compresses the audio signal to match the lip movement. Using this method to provide a stronger baseline to compare Neural Dubbing against would help amplify the value of the proposed method, especially since the included baseline (video-based Tacotron) is  weak and uninformative.

**Quality:**

As stated above, the method is well-motivated, and experiments verify that:
The proposed TTS method does not hurt SOTA TTS performance, as measured by listening study, while controlling the prosody such that the aligned speech is much more synchronized than other methods compared to, also subjectively measured. Objective SyncNet-based metrics also demonstrate superiority over baselines. The supplementary video qualitatively demonstrates some compelling results of the proposed method.

That said, a stronger baseline, as suggested above, would significantly strengthen these claims. Also, in the checklist the authors answered that they did describe the limitations of their work, however, I can't seem to find where this is mentioned. Another experiment that I'm curious about is: how well does this method work when the script is significantly different from what the person is actually saying, e.g., a sentence in another language, or just different words in the same language?

**Clarity:**

The manuscript is clear, and seems to contain enough information to reproduce results. One correction I spotted:
l.154 are suit --> are suitable

**Significance:**

The significance of this work is in the fact that it presents and provides a solution for a novel problem, which is interesting from both a research perspective and from the real-world application viewpoint.


**Time Spent Reviewing:**

3

---

> ### Author Response · Authors · 2021-08-10
> **Response to Reviewer nxuo**
>
> Thanks for your comments on our paper. We reply to your questions as follows:
>
> **[About the missing reference and baseline]**
>
> Thanks for providing the very related reference. We will reference it in the new version of the paper. As you mentioned, our method aims at end-to-end lip-synced TTS (or end2end text2aligned_mel), we need directly generate synchronized speech rather than wrap a pre-recorded speech. So we did not compare with this kind of method before. But it is indeed a good baseline. We will consider using the SOTA TTS method combined with this method as a better baseline in the future.
>
> **[About limitation and the experiment when the script is significantly different from what the person is actually saying]**
>
> As the supplementary video (demo.mp4) shows, our method can only deal with the situation of modifying couple words. In addition, the lip movement of the modified text should be similar to the original lip movement in the video. If the text is modified significantly, the alignment will be very scattered, no longer being a focused diagonal line. This can cause the generated speech to be ambiguous and not intelligible. This is because we use fine-grained attention between lip motion and phonemes.
>
> Please note that silent video dubbing is a task to solve the following application scenario:
>
> > In movie filming, poor sound quality is very common for speech recorded on location. For example, there is obvious noise at the shooting site. In these cases, the speech is re-recorded in a studio during post-production using a process called “Automated Dialogue Replacement (ADR)” or “looping”. In “looping” the actor watches his or her original performance in a loop, and re-performs each line to match the wording and lip movements.
>
> So, in this paper, we only consider the situation where text and lip movement are matched.
>
> **[About the typo]**
>
> Thanks for your correction! We will fix the typo in the new version of the paper.

---

> > ### Comment · Reviewer_nxuo · 2021-08-29
> > **Final comments**
> >
> > In their comments, the authors did a good job of clarifying the type of use-case their system aims to solve, and what is out of scope. Although solving this task end-to-end is novel, it's important that the final version of this manuscript includes the baselines mentioned by the reviewers, namely Lip2wav, and "Dynamic Temporal Alignment", each of which solves the task in a different way (Lip2wav: synced speech, not necessarily matching the text, and "Dynamic Temporal Alignment" which attempts to sync existing (e.g. TTS-generated) speech to lips).

---

> > > ### Author Response · Authors · 2021-08-31
> > > **Response to Reviewer nxuo**
> > >
> > > Thanks for your prompt reply and constructive suggestions! We promise to include the baselines (Lip2Wav and "Dynamic Temporal Alignment") in the final version of the paper. We have already conducted experiments to compare Neural Dubber to SOTA lip-motion based speech generation system Lip2Wav as suggested by Reviewer N6Lx. You can see the experiment details and results in the official comment we post to all titled ***"Some experiment results in the comparison between Lip2Wav and Neural Dubber"***. As the comparison results show, Neural Dubber outperforms Lip2Wav significantly in terms of speech intelligibility, speech quality, and pronunciation accuracy (WER).
> > >
> > > We hope our replies solve all your concerns about the paper. If we successfully address your concerns, we would strongly appreciate an increased score; otherwise, we are happy to provide additional discussion and address any further questions. Thanks!

---

### Official Review · Reviewer_vq7e · 2021-07-16

**Rating:** 6
**Confidence:** 5

**Summary:**

This paper describes a novel task 'silent video dubbing' and proposes a method to solve it. The task involves generating realistic audio to accompany silent video, with the help of text. This is essentially similar to vid2speech [14], but with text guidance. The pipeline consists of several stages. (1) generating phoneme and video embeddings; (2) aligning text and video using the embeddings from (1) and attention network; (3) generating speaker embeddings to condition the decoder from images; (4) using (2) and (3) to generate the result voice. The authors provide example in supplementary materials. The experiments are performed on both single-speaker and multi-speaker SVD.

**Ethical Concerns:**

No concerns

**Limitations And Societal Impact:**

The authors have not addressed this in any detail. They can mention the usual risks associated with deepfake-like research.

**Main Review:**

The task of dubbing silent videos using neural networks is novel to my knowledge. This is an interesting problem in film-making industry.

The pipeline well-engineered, and is well explained. The individual components in the pipeline are fairly standard, rather than novel contributions. However, credit can still be given to how these are put together.

The authors imply that ISE is a contribution of their own, but there are many previous works to learn joint speaker-face embeddings [1,2,3].

I do not think that LSE-D is a fair metric, since this is too similar to the way in which text-video aligner is trained using attention alignment, therefore this unfairly biases towards the proposed method.

Otherwise, qualitative and quantitative results appear to be good.


[1] https://arxiv.org/abs/1805.00833
[2] https://people.csail.mit.edu/changil/assets/face-voice-accv-2018.pdf
[3] https://arxiv.org/abs/2004.14326

**Time Spent Reviewing:**

2hr

---

> ### Author Response · Authors · 2021-08-10
> **Response to Reviewer vq7e**
>
> Thanks for your comments on our paper. We reply to your questions as follows:
>
> **[About the originality of ISE]**
>
> Thank you for providing these references. They are all very good related works, we will reference them in the new version of the paper. I admit that there have been many works researching the correlation between voice and speakers' face. However, none of them (including the references you provide) use the joint speaker-face embeddings to solve the multi-speaker speech synthesis (text to speech) task. Although they learn joint speaker-face embeddings, they can not use them to generate speech with proper/reasonable timbre. In our paper, we learn the image-based speaker embedding (ISE) together with the TTS model without the supervision of speaker identity (actually it is a form of cross-modal self-supervision). And ISE can control the timbre of generated speech based on a speaker's facial image. So I believe ISE is original and is our own contribution.
>
> **[About the LSE-D metric]**
>
> The pretrained SyncNet model we use is from https://github.com/joonson/syncnet_python . It is trained on a totally different dataset based on the paper "Out of time: automated lip sync in the wild". Importantly, Syncnet is trained without any attention module, so it is different to the way in which the text-video aligner is trained. I think there isn't an unfair bias toward our method. Many recent related works have used this metric to evaluate the synchronization of speech and lip motion. In addition, in order to avoid the uncertainty of the SyncNet model, we also conduct human evaluations to judge the audio-visual synchronization and get consistent results.
>
> [1] Out of time: automated lip sync in the wild
> https://www.robots.ox.ac.uk/~vgg/publications/2016/Chung16a/chung16a.pdf

---

> > ### Comment · Reviewer_vq7e · 2021-08-26
> > **Final comments**
> >
> > The authors should include the citations, and add additional comparisons as suggested by Reviewer N6Lx. Otherwise, I am happy for this paper to be accepted.

---

> > > ### Author Response · Authors · 2021-08-31
> > > **Response to Reviewer vq7e**
> > >
> > > Thanks for your prompt reply and constructive suggestions! We will include the citations in the final version of the paper. In addition, we conduct experiments to compare Neural Dubber to SOTA lip-motion based speech generation system Lip2Wav as suggested by Reviewer N6Lx. You can see the experiment details and results in the official comment we post to all titled ***"Some experiment results in the comparison between Lip2Wav and Neural Dubber"***. As the comparison results show, Neural Dubber outperforms Lip2Wav significantly in terms of speech intelligibility, speech quality, and pronunciation accuracy (WER).
> > >
> > > We hope our replies solve all your concerns about the paper. If we successfully address your concerns, we would strongly appreciate an increased score; otherwise, we are happy to provide additional discussion and address any further questions. Thanks!

---

### Official Review · Reviewer_RPCL · 2021-07-17

**Rating:** 7
**Confidence:** 3

**Summary:**

The authors describe the problem of automatic dubbing of videos, ie generating speech from a script that matches the active speaking face in a video. To the best of this reviewer's knowledge, this is the first paper addressing this particular problem. The two main contributions to solve this problem is 1. a method to synchronize the generated TTS with the speaking face 2. sampling the video for a face image to infer a timbre for the speech to better match the face.


**Limitations And Societal Impact:**

The authors might discuss how using a face to influence what the speaking voice sounds like may have some negative impacts or introduce unwelcome bias. What would happen if a female speaker with masculine features were used: would producing a male-sounding voice be offensive? How would race play a role in the sound of the voice?

**Main Review:**

The authors have proposed what seems to be a novel problem and presented a solution which they have evaluated. The problem itself is closely related to a variety of work including [14, 31] where TTS is generated from lip video, without a text constraint (Vid2Speech, Lip2Wav). And Large-scale multilingual audio visual dubbing (https://arxiv.org/pdf/2011.03530.pdf) where the video is alternated to match the generated TTS (translated from the original script) [Yang20].

The evaluation is somewhat flawed in that videos with speech in English have the speech audio re-synthesized. In more practical situations, the dubbed speech is required in a language different to the spoken language in a video. The challenge is coming up with a good  synchronization when the phoneme sequences do not match due to the language mismatch. Still the results are fairly impressive (the demo clips in the appendix are much appreciated).

Originality
The work is original,  although somewhat contrived in that you rarely will have the case that the visemes will match the phonemes well; usually the language of viseme will be different than the phonemes in a new dub. It would be interesting to see how Neural Dubber works in this case.

Clarity
Overall, the paper very nicely introduces the problem which they describe as Silent Video Dubbing.  Perhaps a more accurate term would be Automatic Video Dubbing, since the notion of dubbing already means that the audio track is being replaced with another within the video. SVD is already a common term within the machine learning community, and AVD follows from other similar terms like ASR for automatic speech recognition. The term "silent video" is also awkward; while video can mean media that has both video and audio in the common sense, technically it usually denotes just visual information (https://en.wikipedia.org/wiki/Video) as opposed to audio-visual data. If we don't qualify that video is only visual information, then terms like "Video-based" Tacotron, should be called "Silent video-based" Tacotron, which is strange. In section 4.1 only evaluation datasets are mentioned. Later, some mentions of training data are provided. It would be helpful to include them in section 4.1 to understand what actually needs to be trained and on what corpora.

Quality
The paper clearly motivates this Automatic Video Dubbing problem, presents a solution to guiding the TTS generation to better match the speaking face, and provides a comprehensive analysis of the results with quantitate measurement using AV Sync and qualitative (human ratings). They attempt to provide a good competing baseline system, Video-base Tacotron (although result don't seem so good as the spectrograms don't synchronize well), but the reviewer appreciates the effort. There are results for single speaker and a variety of speakers. The ISE system, is helpful to generate voices that match the speaking faces gender or age, but perhaps can make unfair unbiases on what people should sound like based on their appearance. Still, the paper is comprehensive, and by providing the source code in the appendix, and operating on publicly available data sets, gives experimenters the best chance of being able to reproduce results.

References
The author's have an odd referencing style. There is no need to reference multiple papers for a particular technique, only the first is required. e.g. if [35] is the first example of non-autoregressive TTS, then there is no need to cite [34 ] and [21] as well. Further, [34] is a follow on to [35] (FastSpeech), so unless the paper is citing some thing new in [34], it is better to only cite [35] for anything related to FastSpeech (many times both citations appear together).

Significance
The only downside to the paper, is how the Neural Dubber has been evaluated since re-generating English speech for a video of a person speaking English is a somewhat academic as mentioned earlier (if you already have the English speech, why do you need to re-generate it?). It would be interesting to see how the system would behave would the target phonemes mismatch the viseme language. Unfortunately, this is not answered in this paper. Thus work like in Yang20, is more relevant to the AVD task in practice. The capitalization on the references need to be fixed, e.g. t-sne->T-SNE, FastSpeech, tts -> TTS. Skerry is misspelled in line 449. Please go over the references to make sure they are accurately written.

Minor comments:

line 77: these are not the first papers on TTS. please update.

Figure 1 seems to show that the SVD system doesn't use the green/blue face tracks since they bypass the box, but it fact it does (subtley the video is passed in via the black line). Maybe this could be clearer.

Figure 2. would be helpful to add the labels for S_p, S_n, S_v etc.

line 111: this set of references seemed biased/odd; this is not the complete set of papers evaluating lip reading. it makes more sense to use one that is the most recent state-of-the-art.

line 126-128. the notation is little non-standard. typically lower case letter for scalar, bold lower case for vector and bold upper case for matrix. Thus the phone sequence is \bm{p} = {p_1, p_2 ... and the subscript can mean an index instead of being also overloaded as a indicator of what it is in S_p.

line 150: add that the FFT is simply a transformer with the dense layer replaced with conv. It's an odd/confusing name since the transformer is already feed-forward and FFT usually refers to fast Fourier transform.

Equation (1). the dimensionality of H_vid and H_pho is different, how is this addressed?

Equation (4), the terms are a mixture of quantities e.g. hop size, same rate and units e.g. frame per second. They should all be some quantity e.g. frame rate.

typo line 220

add units in line 239.

Perhaps the Video-based Tacotron would preform better with a constrained alignment process.



**Time Spent Reviewing:**

2

---

> ### Author Response · Authors · 2021-08-10
> **Response to Reviewer RPCL**
>
> Thanks for your comments on our paper. We reply to your questions as follows:
>
> **[About using the method of re-synthesizing speech for evaluation]**
>
> There are two common application scenarios for dubbing. The first application scenario is: In movie filming, poor sound quality is very common for speech recorded on location. Maybe there is obvious noise at the shooting site, or the scene itself was too challenging to record high-quality audio. In these cases, the speech is re-recorded in a studio during post-production using a process called “Automated Dialogue Replacement (ADR)” or “looping”. In “looping” the actor watches his or her original performance in a loop, and re-performs each line to match the wording and lip movements.
>
> The second application scenario is translating foreign-language films into the audience’s language. For example, the original English video is translated into Chinese.
>
> In this paper, we mainly focus on the first application scenario, where the lip motion (viseme) in the video is consistent with the given text. The reason why we need to re-generate speech is that, in a ADR scenario, the original audio in the video may be very noisy or the actor’s voice is not recorded. This requires our method to generate high-quality, synchronized speech based on the lines (scripts) and the video, so we use the method of re-synthesizing speech for evaluation.
>
>
> **[About changing Silent Video Dubbing (SVD) to Automatic Video Dubbing (AVD)]**
>
> Thanks for your suggestion. Automatic Video Dubbing (AVD) is a very good name. We originally used Silent Video Dubbing (SVD) because we wanted to emphasize that we only need visual information and lines to dub (re-record) films during post-production process called “Automated Dialogue Replacement (ADR)”. We will change the name in the new version of the paper.
>
> **[About the evaluation datasets mentioned in section 4.1]**
>
> "In the single-speaker setting, we evaluate Neural Dubber on the chemistry lecture dataset from Lip2Wav [31]." The word "evaluation" here actually means doing experiments, including training, validation and testing. So "In the single-speaker setting, we conduct experiments on the chemistry lecture dataset from Lip2Wav [31]." is clearer. We will fix it in the new version of the paper.
>
> **[About referencing style]**
>
> Thanks for your suggestion. We are used to reference multiple well-known papers for a particular topic or technique. We will follow your suggestions to check the references carefully and fix it in the new version of the paper.
>
> **[About the question on Figure 1]**
>
> In Neural Dubber, we use the green/blue face tracks in fact. The green/blue lines bypass the SVD block to express that the synthesized speech is synchronized to the active speaker's lip motion in the video. We will add more captions to make it clear in the new version of the paper.
>
> **[About the question: the dimensionality of H_vid and H_pho is different in Equation (1)]**
>
> Actually, the dimensionality of H_vid and H_pho are the same. You can see Line 133 and 134.
>
> **[About typos, notations and quantities, and other minor comments]**
>
> Thanks for your advice! We fix them in the new version of the paper.

---

> ### Author Response · Authors · 2021-08-31
> **Update reminder for Reviewer RPCL**
>
> Thanks for your constructive comments on our paper. Recently, we conduct experiments to compare Neural Dubber to SOTA lip-motion based speech generation system **Lip2Wav** as suggested by Reviewer N6Lx. We think the results of this experiment will also be useful to your judgment. So we suggest that you can also check the results. You can see the experiment details and results in the official comment we post to all titled ***"Some experiment results in the comparison between Lip2Wav and Neural Dubber"***. As the comparison results show, Neural Dubber outperforms Lip2Wav significantly in terms of speech intelligibility, speech quality, and pronunciation accuracy (WER).
>
> We hope the experiment results are also helpful to solve your concerns about the paper. If we successfully address your concerns, we would strongly appreciate an increased score; otherwise, we are happy to provide additional discussion and address any further questions. Thanks!

---

### Official Review · Reviewer_N6Lx · 2021-07-19

**Rating:** 7
**Confidence:** 4

**Summary:**

In this paper, the authors propose to solve the problem of text to speech generation based on video. Thus, this is video-text guided speech generation task that they term as silent video dubbing. The method is based on a transformer architecture that combines text and visual lip motion representations as encoders and outputs the mel-spectrograms through the decoder of a transformer.

The method is compared with TTS systems and comparisons are done using human evaluation scores and audio-visual synchronisation scores. This is done over LRS2 and Lip2Wav for one out of 5 speakers. Comparisons are not done using standard PESQ, STOI or Word error rates. These are standard metrics used for TTS systems. Comparison is also not provided with single speaker Lip2Wav task or multi-speaker Lip2Wav on LRW dataset as done by Lip2Wav or Vid2Speech, or such methods that generate speech based on lip motion alone (not requiring text).

**Ethical Concerns:**

The authors acknowledge that the ethical issues are not fully addressed. Presently it uses only publicly available data sources. Thus, the issues are not different to existing similar systems. This can be acknowledged.

**Limitations And Societal Impact:**

The submission can include possible societal concerns, though its application for creative industries can be useful.

**Main Review:**

The method aims to solve the task of `silent video dubbing’ where given a video and a text, the aim is to generate the corresponding speech that has the same prosody as the video, thereby enabling the task of dubbing.

The method is based on a text-video alignment based transformer that additionally uses an image specific embedding with a variance adaptor (used in FastSpeech2) to match pitch and energy for the generated mel-spectrograms generated by the decoder. As is common in transformer architectures, positional encoding blocks are included.  The proposed architecture is modified from existing TTS systems to include lip video embeddings as an additional cue.


Pros:
The problem aims to address the generation of speech conditioned on text and video as compared to existing methods that aim to generate speech based only on text or generation of speech based only on lip motion.

The proposed method shows evaluation on LRS2 dataset and evaluation for a single speaker from a speaker-specific dataset. The evaluation is done using MOS scores, evaluation of a speaker specific speech generation visualisation and lip-synchronisation accuracy. The comparisons are done with TTS systems.


Cons:

Evaluation limitations:

The main limitation of this work is that while, the novel task aims to combine the ability to generate speech from text and lip-motion, the comparisons are provided only with TTS systems and not with lip-motion based speech generation systems. Clearly, if the method uses more information as compared to lip-motion based speech generation systems, it would be relevant to compare their output with such systems

The second limitation of this work is that the evaluation measures are obtained through human evaluation and synchronisation, however, the standard metrics used for TTS systems such as PESQ, STOI are not evaluated. This is important as the quality of speech generation is very crucial for dubbing applications (as proposed in the paper). The overall qualitative comparison provided suggests that the actual audio quality is somewhat inferior to TTS systems, though the synchronisation appears to be better.

Similarly, when evaluating on single speaker setting, it would have been easy to provide a comparison with Lip2Wav as the setting is similar (only more information is added). The lip-motion based systems have also been compared using PESQ and STOI metrics appear to provide better audio as compared to the proposed system (quantitative comparison is not possible due to lack of results through the metrics).

This implies that though the proposed method uses more parameters (transformer architecture) and more information, the quality appears to be lesser as compared to lip-motion based generation and text-to-speech based systems.


Minor writing issues:

While, on the whole the paper is well written and clear, the paper has a few minor grammatical/spelling errors. The line numbers for the same are provided:
38 “an one-to-many”
73 “in a big margin”
108  “quailty gains”
127 “an phoneme sequence”
154 “are suit for capturing”

Conclusion:
To conclude, while the paper proposes an interesting variant, due to the limitations in terms of evaluation, I tend to be negative about this work.



**Time Spent Reviewing:**

4 hours

---

> ### Author Response · Authors · 2021-08-10
> **Response to Reviewer N6Lx**
>
> Thanks for your comments on our paper. We reply to your questions as follows:
>
> **[About not comparing with lip-motion based speech generation systems]**
>
> Thanks for your advice, we will include a comparison with Lip2Wav in the final version of the paper. Because our silent video dubbing (SVD) task is essentially derived from text to speech task, it just uses the lip motion in the video to control the prosody of generated speech. And Neural Dubber is a multi-modal TTS model, aims at the end-to-end text to aligned mel-spectrograms generation. The intuition of lip-motion based methods is different from ours, so we did not compare our method with such methods. Besides, recently SOTA lip-motion based speech generation system, like Lip2Wav, can only generate word-level speech for multi-speaker. In addition, the audio-visual synchronization (AV sync) is very important for SVD task, but LRW dataset only contains clips of a single word (too short),  thus AV sync cannot be well tested on LRW dataset, and that's why we didn't use LRW dataset for SVD task.
>
>
> **[About not using PESQ, STOI and Word error rate]**
>
> Actually, PESQ, STOI and Word error rate are not be used in recent TTS papers. Many well-known TTS papers[1,2,3,4,5] don't use these metrics. The most commonly used evaluation metric in the TTS field is Mean Opinion Score (MOS), which is also used in our paper.
>
>
> [1] TACOTRON: TOWARDS END-TO-END SPEECH SYNTHESIS
>
> [2] NATURAL TTS SYNTHESIS BY CONDITIONING WAVENET ON MEL SPECTROGRAM PREDICTIONS
>
> [3] FastSpeech: Fast, Robust and Controllable Text to Speech
>
> [4] FASTSPEECH 2: FAST AND HIGH-QUALITY END-TO-END TEXT TO SPEECH
>
> [5] WAVENET: A GENERATIVE MODEL FOR RAW AUDIO
>
> **[About misunderstanding that our audio quality is inferior to TTS systems]**
>
> Actually, our audio quality is better than that of TTS systems. As you can see from Table 1 (or 2), the MOS result of audio quality is Neural Dubber (Mel + PWG) 3.74 (3.58) vs FastSpeech 2 (Mel +PWG) 3.71 (3.15). The results show that Neural Dubber is better than FastSpeech 2 in term of speech quality both in single and multi speaker settings.
>
> **[About misunderstanding that the lip-motion based systems provide better audio as compared to the proposed system]**
>
> Neural Dubber can not be compared with Lip2Wav directly, because the chemistry lectures dataset from Lip2Wav we used is different. We use different data cleaning methods: first, we segment the videos into short clips according to the start and end timestamp of each sentence in the transcripts; second, we remove sentence-level clips which exist a frame where the lecturer is absent. So after cleaning, there are only about 9 hours video clips left for originally about 24 hours of data. Compared to Lip2Wav, the number of our data is less, so you can’t directly compare their performance based on the audios in supplementary materials. Later we will train Lip2Wav on our data for fair comparison.
>
> **[About model parameters and method comparison]**
>
> Lip2Wav uses a stack of 3D convolutions as video encoder, while Neural Dubber only uses 2D convolutions. The parameter of Lip2Wav on the video encoder is much larger than that of Neural Dubber. Because the transformer model we use is relatively small, even if the total parameters of the models are compared, Neural Dubber will not be larger than Lip2Wav.
>
> In fact, our model is more powerful than the lip-motion based method. The SOTA lip-motion based method Lip2Wav can only generate word-level speech for multi-speaker, while Neural Dubber can generate high quality sentence-level speech for multi-speaker with proper timbre, and generate different speech by changing couple words in the text (please see our demo.mp4 in supplementary materials).
>
> The novel task SVD is harder than TTS. We need to generate high-quality and lip-synced speech from the given text and silent video. This requirement is obviously much more difficult than TTS. Neural Dubber can solve SVD well in both single-speaker and multi-speaker settings. Experiments show that Neural Dubber outperforms TTS model obviously.
> So I think this is enough to prove our contribution and novelty.
>
> **[About the typo]**
>
> Thanks for your correction! We will fix the typo in the new version of the paper.

---

> > ### Comment · Reviewer_N6Lx · 2021-08-25
> > **Response**
> >
> > Thanks to the authors for their response.
> >
> > 1) Comparison with lip-motion based speech generation systems: I am still not fully convinced by the argument. For instance, if the intention for the authors is to use it for movie dubbing, clearly there are sufficient speaker specific content available and in this setting Lip2Wav better. At least a comparison in this setting would be warranted. The authors promise to compare with Lip2Wav in the final version and it would be useful to see such a comparison. The limitations of Lip2Wav for multi-speaker setting are known. However, the audio response from the proposed system is actually not yet in a stage that it can be used. Specially in settings such as ISE ablation setting it is clearly that the output is not yet convincing. The main limitation is that this setting actually has more information than only lip-based audio generation as both video and text are available at input. In this setting the output should be clearly better.
> >
> > 2) The authors argue that PESQ, STOI and such metrics are not required and MOS suffices. However, in usage of dubbing in actual practice, it is very important to validate the audio quality. Papers such as FastSpeech2 do validate their works using other metrics such as MAE (Mean absolute error), statistics of DTW for pitch and such comparisons. Using LipSync based metrics can validate the sync but cannot validate the output quality. Some measure based on this would definitely be quite essential for the proposed task. The comparison of only MOS scores does not suffice to convince about the audio quality, particularly when one listens to the supplementary video for the multi-speaker audio output. I am actually not fully convinced that the audio improvement can stand out, especially given the output from other systems such as Lip2Wav.
> >
> > 3) Thank you for clarifying on the differences from Lip2Wav dataset. It is not clear why exactly this difference was adopted. Lip2Wav also uses other individuals. It would be interesting to consider the output for the other speakers also. It is known that the Chemistry speaker is one of the easiest to get good results on. The other speakers are more challenging. These were not compared nor output shown for these other speakers.
> >
> > 4) Thanks for clarifying the parameter and model differences. The advantage of the proposed method over TTS is well understood. Similarly the advantage over multi-speaker Lip2Wav is also well understood. The only concern is that in speaker specific setting, the model should be expected to far outperform speaker specific Lip2Wav like models. This is because the proposed method has more inputs and is an improved architecture. This improvement is necessary to validate that actual progress is achieved.
> >
> > In view of these, I continue to have concerns about the paper. However, given that the authors promise to compare it with speaker specific Lip2Wav I am willing to raise my rating from 4 to 5.

---

> > > ### Author Response · Authors · 2021-08-30
> > > **Response to Reviewer N6Lx**
> > >
> > > Thanks for your prompt reply and for raising your rating. To further solve your concerns, we conduct experiments to compare Neural Dubber to SOTA lip-motion based speech generation system Lip2Wav. **You can see the experiment details and results in the official comment we post to all.** The following is the response to your questions.
> > >
> > >
> > >
> > > **[About what model is suitable for movie dubbing]**
> > >
> > > As the comparison results show, Neural Dubber outperforms Lip2Wav significantly in terms of speech intelligibility, quality, and pronunciation accuracy (WER). Especially the WER of Lip2Wav is quite high, which is totally unsuitable for the task of film dubbing. It is unacceptable for an actor/actress to always mispronounce the lines. Moreover, in the scenario of film dubbing, text (lines) and video are readily available. So it is very reasonable to use them all to solve the video dubbing task. In summary, Neural Dubber is more suitable for movie dubbing.
> > >
> > > **[About the reason why the speech quality in the multi-speaker setting is not good enough]**
> > >
> > > The LRS2 dataset is originally used for lip reading task which is a recognition task rather than a generation task. The audio quality of the LRS2 dataset is poor, the background noise is large. In addition, there are too many speakers, and the time each person speaks varies greatly. These problems of the dataset lead to poor quality of the synthesized speech. The ISE can make Neural Dubber generate speech with reasonable timbre to further improve the perceptual quality.
> > >
> > > **[Respose to Question 2 and 4]**
> > >
> > > As the above comparison results show, Neural Dubber outperforms Lip2Wav significantly under the STOI, ESTOI, PESQ, and WER metrics. These results demonstrate that Neural Dubber far outperforms Lip2Wav in terms of speech intelligibility, quality, and pronunciation accuracy (WER) in the speaker specific setting. These improvements validate that actual progress is achieved.
> > >
> > > **[About the reason why the difference of the chemistry dataset was adopted]**
> > >
> > > Neural Dubber is a multi-modal TTS model. In the TTS task, we need corresponding sentence-level text and audio clips. So we need to segment the long videos into sentence-level clips according to the start and end timestamp of each sentence in the transcripts. There are many video sections containing only pictures (PPT) and no lecturer face in the chemistry lecture dataset, which causes some frames in some segmented sentence-level video clips that only contain PPT but not lecturer face. This kind of sentence-level video clip cannot be used for training. So we need to conduct data cleaning to remove sentence-level clips which exist frames where the lecturer is absent. Finally, the dataset contains 6,640 sentence-level video clips with a total video length of approximately 9 hours.
> > >
> > > **[About not using other speaker specific datasets]**
> > >
> > > We just want to show that our model can handle the single-speaker setting for the video dubbing task, so we chose one dataset randomly from those which have official subtitles on YouTube. There is no essential difference between other speakers and the Chemistry speaker, except that the image area of other speakers is small and they have accents. But the Chemistry speaker has more head motion which makes it challenging. So we believe that Neural Dubber can also handle datasets of other speakers.
> > >
> > >
> > >
> > > We hope our replies solve all your concerns about the paper. If we successfully address your concerns, we would strongly appreciate an increased score; otherwise, we are happy to provide additional discussion and address any further questions. Thanks!

---

### Author Response · Authors · 2021-08-10
**Response to All**

**[About the application scenario of the Silent Video Dubbing task]**

There are two common application scenarios for dubbing.

The first application scenario is: In movie filming, poor sound quality is very common for speech recorded on location. Maybe there is obvious noise at the shooting site, or the scene itself was too challenging to record high-quality audio. In these cases, the speech is re-recorded in a studio during post-production using a process called “Automated Dialogue Replacement (ADR)” or “looping”. In “looping” the actor watches his or her original performance in a loop, and re-performs each line to match the wording and lip movements.

The second application scenario is translating foreign-language films into the audience’s language. For example, the original English video is translated into Chinese.

In this paper, we mainly focus on the **first** application scenario, where the lip motion (viseme) in the video is consistent with the given text (phoneme). The reason why we need to re-generate speech is that, in a ADR scenario, the original audio in the video may be very noisy or the actor’s voice is not recorded clearly. This requires our method to generate high-quality, synchronized speech based on the lines (scripts) and the visual information.


**[Limitations And Societal Impact]**

Limitation:
1. Only couple of words are allowed to be altered, so our method may not generalize well to translate film scenarios where lip motion and text are totally mismatched.
2. Dubbing for speech translation is not able to be achieved now since the lack of a parallel multi-lingual video dataset. So we left this task as future work.

Societal impact:
1. The facial appearance may lead to timbre ambiguity due to the dataset bias. For instance, actually, the pitch of a man could be as high as a woman, although he has a masculine face. But our model will produce a low male voice. It might be offensive.
2. Since our method can generate different speech by altering the words of given scripts, it may bring some social damages, such as video fraud.

---

### Author Response · Authors · 2021-08-30
**Some experiment results in the comparison between Lip2Wav and Neural Dubber**

**[Comparison between Lip2Wav and Neural Dubber in single-speaker setting]**

Because Lip2Wav can only generate word-level speech in the multi-speaker setting. We compare Neural Dubber to SOTA lip-motion based speech generation system Lip2Wav in the single-speaker setting still on the chemistry lecture dataset. We use the official GitHub repo https://github.com/Rudrabha/Lip2Wav/ to train Lip2Wav on our version of the chemistry lecture dataset. As we mentioned in the paper and last reply, the dataset is different from the original one in Lip2Wav. It only contains data of approximately 9 hours, which is much less than the original one (approximately 24 hours). In this experiment, the training and testing sets of Neural Dubber and Lip2Wav are identical, so the results can be compared directly now. Following the Lip2Wav paper[1], we use STOI and ESTOI for estimating the intelligibility and PESQ for measuring the quality. In addition, using an out-of-the-box ASR system, we can evaluate the speech results using word error rates (WER), because we obtain the ground truth text transcripts from YouTube official subtitles. In order to eliminate the influence of the ASR system, we obtain textual transcripts for ground truth speech audio, and evaluate the results using WER compared to ground truth text transcripts. All these metrics are computed on the test dataset.



| Method | STOI | ESTOI |PESQ| WER|
| :-- | :-----: | :-----: | :-----: | :-----: |
| Ground Truth |  NA  | NA  | NA | 7.57%|
| Lip2Wav | 0.282 |  0.176  |  1.194  | 72.70% |
| **Neural Dubber (ours)** | **0.467** |  **0.308**  |  **1.250**  | **18.01%** |

Table 1: The comparison between Lip2Wav and Neural Dubber for the single-speaker setting on the chemistry lecture dataset.



As the comparison results in Table 1 show, Neural Dubber surpasses Lip2Wav by a big margin in terms of speech quality and intelligibility. The scores of STOI, ESTOI, and PESQ metrics of Neural Dubber are much higher than those of Lip2Wav. Please note that STOI, ESTOI, and PESQ scores of Lip2Wav are lower than those in the Lip2Wav paper[1], because the training data is much less than the original one. Most importantly, the WER of Neural Dubber is 4x lower than that of Lip2Wav. It shows that Neural Dubber outperforms Lip2Wav significantly in pronunciation accuracy. WER of Lip2Wav is up to 72.70%, indicating that it mispronounces a lot of content, which is unacceptable in the task of video dubbing. Just like it is unacceptable for an actor/actress to always mispronounce the lines. Please note that the WER of Lip2Wav we get is consistent with the results of the Lip2Wav paper (see Table 5)[1]. In summary, Neural Dubber far outperforms Lip2Wav in terms of speech intelligibility, quality, and pronunciation accuracy (WER), and is much more suitable for the video dubbing task.


It is also worth pointing out that Neural Dubber is a **non-autoregressive** model, which generates mel-spectrograms **in parallel**; however, Lip2Wav is an autoregressive model, which generates mel-spectrograms in sequence. As a result, Neural Dubber is **significantly faster** than Lip2Wav in terms of training and inference speed. Just like the comparison between FastSpeech / FastSpeech 2 and Tacotron / Tacotron 2.


[1] Learning Individual Speaking Styles for Accurate Lip to Speech Synthesis https://arxiv.org/pdf/2005.08209.pdf




**[The reason why the difference of the chemistry dataset was adopted]**

Neural Dubber is a multi-modal TTS model. In the TTS task, we need corresponding sentence-level text and audio clips. So we need to segment the long videos into sentence-level clips according to the start and end timestamp of each sentence in the transcripts. There are many video sections containing only pictures (PPT) and no lecturer face in the chemistry lecture dataset, which causes some frames in some segmented sentence-level video clips that only contain PPT but not lecturer face. This kind of sentence-level video clip cannot be used for training. So we need to conduct data cleaning to remove sentence-level clips which exist frames where the lecturer is absent. Finally, the dataset contains 6,640 sentence-level video clips with a total video length of approximately 9 hours.

---

### Decision · Program_Chairs · 2021-09-27

**Decision:**

Accept (Poster)

**Comment:**

The original submission was missing an essential comparison with Lip2Wav. This has been added in the rebuttal period, and all reviewers now recommend acceptance.

The authors should include the Lip2Wav comparison and the missing references highlighted by the reviewers in the final version.